# Activity data from wearables as an indicator of functional capacity in patients with cardiovascular disease

Neil Rens[1,2]ʘ*, Neil Gandhi[1,2]ʘ, Jonathan Mak[1‡], Jeddeo Paul[1,2‡], Drew Bent[2], Stephanie Liu[2], Dasha Savage[1], Helle Nielsen-Bowles[1], Doran Triggs[1], Ghausia Ata[1], Julia Talgo[1], Santiago Gutierrez[1], Oliver Aalami[1]

1 Division of Vascular Surgery, Department of Surgery, Stanford University, Stanford, California, United States of America, 2 Graduate School of Business, Stanford University, Stanford, California, United States of America

ʘ These authors contributed equally to this work.
‡ These authors also contributed equally to this work.
* neilerens@gmail.com

## Abstract

### Background

Smartphone and wearable-based activity data provide an opportunity to remotely monitor functional capacity in patients. In this study, we assessed the ability of a home-based 6-minute walk test (6MWT) as well as passively collected activity data to supplement or even replace the in-clinic 6MWTs in patients with cardiovascular disease.

### Methods

We enrolled 110 participants who were scheduled for vascular or cardiac procedures. Each participant was supplied with an iPhone and an Apple Watch running the VascTrac research app and was followed for 6 months. Supervised 6MWTs were performed during clinic visits at scheduled intervals. Weekly at-home 6MWTs were performed via the VascTrac app. The app passively collected activity data such as daily step counts. Logistic regression with forward feature selection was used to assess at-home 6MWT and passive data as predictors for "frailty" as measured by the gold-standard supervised 6MWT. Frailty was defined as walking <300m on an in-clinic 6MWT.

### Results

Under a supervised in-clinic setting, the smartphone and Apple Watch with the VascTrac app were able to accurately assess 'frailty' with sensitivity of 90% and specificity of 85%. Outside the clinic in an unsupervised setting, the home-based 6MWT is 83% sensitive and 60% specific in assessing "frailty." Passive data collected at home were nearly as accurate at predicting frailty on a clinic-based 6MWT as was a home-based 6MWT, with area under curve (AUC) of 0.643 and 0.704, respectively.

**Data Availability Statement:** All relevant data are within the manuscript and its Supporting Information files.

**Funding:** The study received funding from Apple, Inc. and the Precision Health and Integrated Diagnostics Center at Stanford. None of the funding sources impacted the study design, data collection, analysis, decision to publish, or preparation of this manuscript. The specific roles of these authors are articulated in the 'author contributions' section.

**Competing interests:** The authors have read the journal's policy and declare the following competing interests: Apple, Inc provided support via internship salary for SG. There are no patents, products in development or marketed products associated with this research to declare. This does not alter our adherence to PLOS ONE policies on sharing data and materials.

## Conclusions

In this longitudinal observational study, passive activity data acquired by an iPhone and Apple Watch were an accurate predictor of in-clinic 6MWT performance. This finding suggests that frailty and functional capacity could be monitored and evaluated remotely in patients with cardiovascular disease, enabling safer and higher resolution monitoring of patients.

## Introduction

Peripheral arterial disease (PAD), valvular disease, and coronary artery disease (CAD) affect over 10 million, 6 million, and 18 million people in the United States, respectively [1–3]. CAD alone accounted for nearly 366 thousand deaths in 2017 and 11 million clinic visits [3]. Prognosis and clinical status for patients with these diseases can be assessed using the clinic-based six-minute walk test (6MWT) [4]. Ability to remotely capture the same data as a clinic-based 6MWT could increase opportunities for remote management of patients with cardiovascular disease (CVD).

With remote management, patients can receive care from the comfort of their homes, saving time and reducing transportation hurdles. Continuous remote monitoring also creates opportunities for more continuous rather than episodic care, allowing for more personalized and timely interventions [5–7]. The adoption of remote monitoring was already on the rise prior to COVID, and COVID has accelerated its adoption [8–10].

Smartphones with embedded accelerometers allow assessment of a person's functional capacity through passive accelerometer data analysis. For patients with CVD, functional capacity is increasingly relevant for assessing frailty and quality-of-life status [11], making widespread adoption of smartphones an unprecedented opportunity for remote monitoring of CVD. However, since most step and distance count algorithms were developed with younger and fitter users in mind, validation of data collection in patient populations with CVD is required [12]. In our study of patients with CVD, we assessed the ability of home-based 6MWT as well as passively collected activity data to supplement or even replace the in-clinic 6MWTs over the course of their surgical interventions.

## Methods

The data that support the findings of this study are provided in the supporting information files. The study was registered with ClinicalTrials.gov with identifier number NCT03048890. The study received approval from the Stanford University Institutional Review Board.

### Study design

Ambulatory patients with CAD who were scheduled for coronary artery bypass graft (CABG) or percutaneous coronary intervention (PCI), patients with valvular insufficiency scheduled for mitral valve replacement (MVR) or transaortic valve replacement (TAVR), and patients with peripheral vascular disease scheduled for vascular bypass or endovascular procedures were enrolled into our IRB approved study at the Palo Alto Veterans Affairs Hospital. Patients were provided with an iPhone 7 and Apple Watch Series 3 loaded with the VascTrac research application developed by our group (https://apps.apple.com/us/app/vasctrac/id1121791155). Patients were also supplied a data plan for the duration of the study. Patients were consented

and onboarded with the support of a clinical research coordinator prior to their planned intervention and performed an in-clinic, supervised 6MWT per the VascTrac study protocol [12]. Patients received weekly push notifications on their iPhone to perform at-home 6MWTs.

## Data collected

The VascTrac research application collected: 1) total steps taken each day, 2) the number of steps walked without stopping for more than a minute, i.e. maximum steps without stopping (MSWS), and 3) total distance walked per day. These metrics were passively collected, meaning they were recorded continuously while the participant had either the smartphone or wearable on them and did not require any involvement from the participant. Patients repeated the in-clinic supervised 6MWT during their post-procedure follow-up visits at 2 weeks and at 1, 3, and 6 months. On weeks when patients did not have a clinic appointment for the study, they completed a 6MWT at home using their phone and watch. The study duration was 6 months.

## Accuracy of smartphone-based 6MWT in clinic

In existing literature, a patient is defined as 'frail' if she or he achieves less than 300 meters on a 6MWT [13–15]. Frailty is a validated marker for poor outcomes and thus we used it to assess the accuracy of the smartphone-based 6MWT. For a given clinic-based 6MWT, we assessed the correlation between iPhone distance and ground-truth distance (measured by a clinical coordinator) using a linear regression. We also assessed the correlation between iPhone step count and ground-truth distance. We then performed logistic regressions to compare the accuracy of these two smartphone data sources in predicting whether a patient would be classified as "frail" based on their ground-truth distance.

## Using smartphone data from home to predict clinic 6MWT performance

We compared the ability of two different data streams—passive activity data and a home-based 6MWT—to predict performance on the participant's next in-clinic 6MWT. We created a logistic regression using features from each data stream (see S1 Appendix for full list of features tested), with "frailty" as the binary outcome.

For the home-based 6MWT prediction, we identified the at-home 6MWTs that occurred nearest to the next in-clinic 6MWTs (median: 6 days) for every given patient and calculated various features, which included variations of step counts taken from both the phone and the watch. The 110 patients were divided 60% into training data and 40% into testing data, resulting in 229 unique training 6MWTs and 151 testing 6MWTs. The logistic regression model was trained using a 5-fold cross validation repeated 10 times with a binary outcome of frailty defined by 300 meters. The best model was determined by forward selection comparing the mean validation area under the curve (AUC) of each model across all 50 runs [16]. To determine a 95% confidence interval of each model's AUC during the testing phase, we performed a bootstrapped analysis on 200 training sets and 1000 testing sets (see S1 Appendix).

The same forward selection model was used to predict frailty using passive activity data. The model's inputs were the 7 days of passive data immediately preceding each clinic-based 6MWT. The set of all features extracted from passive data are shown in the (S2 Fig in S1 Appendix).

## Results

### Study participants

One hundred and fifty-six (156) patients were invited to participate during cardiac surgery or vascular surgery clinic visits, and 110 patients were enrolled between May 2018 and May 2019. There were 109 male participants (99%—due to this study being implemented at a VA hospital) with a mean age of 68.9 years (ranging from 57–89 years). The average BMI was 28.8 kg/m$^2$. Fifty-six patients (51%) were former smokers and 33 patients (30%) were current smokers. Ninety-four patients (85%) participants had hypertension, 39 (35%) had diabetes mellitus, 23 (21%) had aortic stenosis, 16 (15%) had atrial fibrillation, and 4 (4%) had heart failure. More patient characteristics and medication profiles can be seen in Table 1.

Patients underwent a total of 101 procedures. There were 59 peripheral arterial procedures (18 open bypass and/or endarterectomies and 41 endovascular procedures). There were 42 cardiac procedures (28 CABG/AVRs, 6 TAVRs, 5 PCIs and 3 MVRs).

Out of the 110 study participants, there were 16 (15%) premature exits. There were 3 (3%) deaths prior to completing the study, 7 (7%) chose to withdraw, and 6 (5%) were lost to follow up. Of the 7 who chose to withdraw, 4 were due to medical complications, 1 lost his devices, and 2 patients moved out of state.

In total, patients performed 450 supervised in-clinic 6MWTs and 2,019 home 6MWTs (Fig 1). Across all patients, 58 million steps and 5,144 flights of stairs were catalogued.

**Table 1. Demographic characteristics of study participants.**

| Physical Characteristics | Mean +/- SD n (%) | Comorbidities | n (%) |
|---|---|---|---|
| Sex (male) | 109 (99%) | PAD | 69 (63%) |
| Age | 68.9 +/- 5.9 | CAD | 61 (57%) |
| Height (in) | 69.0 +/- 3.4 | Diabetes Mellitus | 39 (35%) |
| Weight (lbs) | 195.3 +/- 40.4 | Hypertension | 94 (85%) |
| BMI (mean) | 28.8 +/- 4.7 | Aortic Stenosis | 23 (21%) |
| BMI >30 | 40 (36%) | Atrial Fibrillation | 16 (15%) |
| Current Smoker | 33 (30%) | Mitral Stenosis | 2 (2%) |
| Former Smoker | 56 (51%) | Mitral Regurg. | 16 (15%) |
| Never Smoked | 12 (11%) | CHF | 4 (3.9%) |
| Spouse/ Caregiver | 73 (66%) | History of MI | 4 (4%) |
| Smartphone Naive | 25 (23%) | ESRD | 2 (2%) |
| **Medications** | **n (%)** | **Total Interventions** | **n (%)** |
| Aspirin | 92 (84%) | **Peripheral Arterial** | **59 (58.4%)** |
| Plavix/Clopidogrel | 52 (47%) | PAD endovascular | 41 (40.6%) |
| Statins | 96 (87%) | PAD Open | 18 (17.8%) |
| Insulin | 16 (15%) | | |
| Warfarin | 13 (12%) | **Cardiac** | **42 (41.6%)** |
| NOACs | 11 (10%) | CABG/AVR | 28 (27.7%) |
| **Ankle Brachial Index (ABI)** | **Mean +/- SD** | TAVR | 6 (5.9%) |
| Baseline ABI | 0.73 +/- 0.26 | PCI | 5 (4.9%) |
| Post-Op ABI | 0.88 +/- 0.23 | MVR | 3 (2.9%) |
| **Ejection Fraction** | **% +/- SD** | | |
| Baseline EF (%) | 55 +/- 10.3 | | |
| Post-Op EF (%) | 56 +/- 8.5 | | |

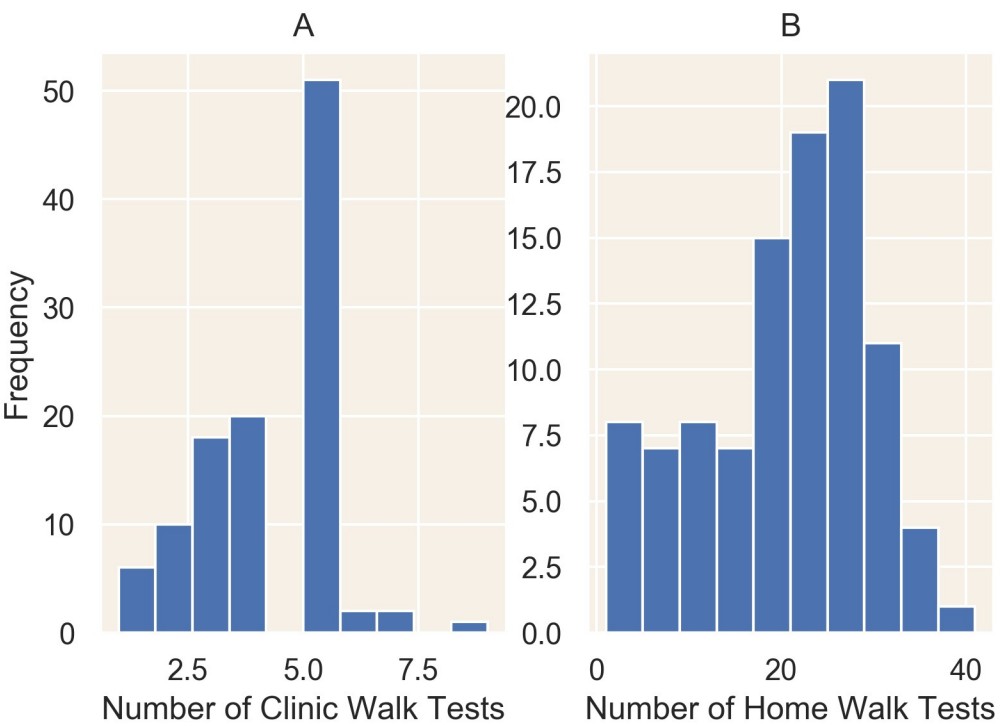

**Fig 1. Frequency of clinic-based (A) and home-based (B) six-minute walk tests.** Patients averaged 4 clinic-based walk tests, with most patients completing at least 6 (A). Only 15% completed two or fewer. Patients averaged 20 home-based walk tests, with 82% of patients completing at least 10 home-based walk tests (B).

### Accuracy of smartphone-based 6MWT in clinic

Steps captured by the smartphone correlated well with the in-clinic ground truth 6MWT distance. The steps did not correlate as well with the distance captured by the smartphone (Fig 2A and 2B, correlation coefficients of 0.84 and 0.78, respectively). Likewise, smartphone steps outperformed smartphone distance when predicting frailty using a logistic regression (AUC 0.919 vs 0.885, respectively). However, the smartphone was most accurate in classifying a walk test as "frail" when using both smartphone steps and smartphone distance (Fig 2C, AUC 0.924).

### Using smartphone data from home to predict clinic 6MWT performance

The logistic regression using the home-based 6MWT performed best with a single feature: natural logarithm of the steps counted by the phone or watch, whichever was higher. With this feature, the model achieved an AUC of 0.704 (Fig 3A). The logistic regression using the passive activity data performed best with a single feature: natural logarithm of the total steps walked per day as counted by the phone or watch, whichever was higher. With this feature, the model achieved an AUC of 0.643 (Fig 3B). Both models selected just one feature each, despite having access to multiple features (see S1 Appendix for full list of features tested).

### Discussion

Functional capacity is increasingly an important metric for patient-centered care for the CVD population [11]. While many studies have examined 6MWTs in the CVD population, few have examined a self-administered smartphone-based version despite the increasing need for

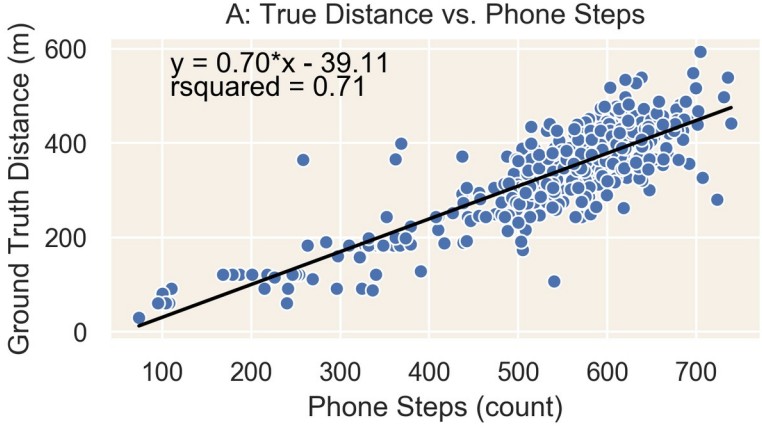

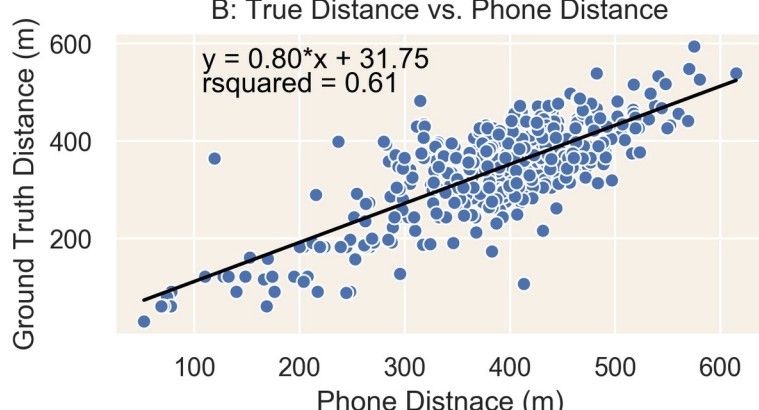

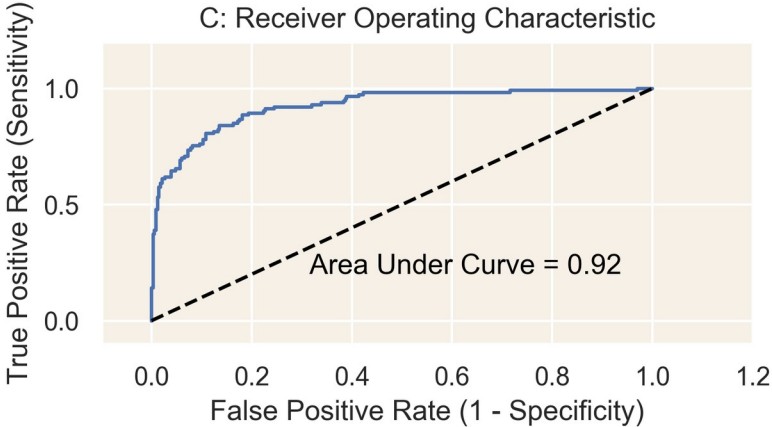

**Fig 2. Accuracy of smartphone based six-minute walk test.** For clinic-based six-minute walk tests, the smartphone step counts were more highly correlated with ground truth distance than was the smartphone-measured distance. When used to detect frailty as measured by ground truth distance <300m, the smartphone performed best when incorporating both step and distance data.

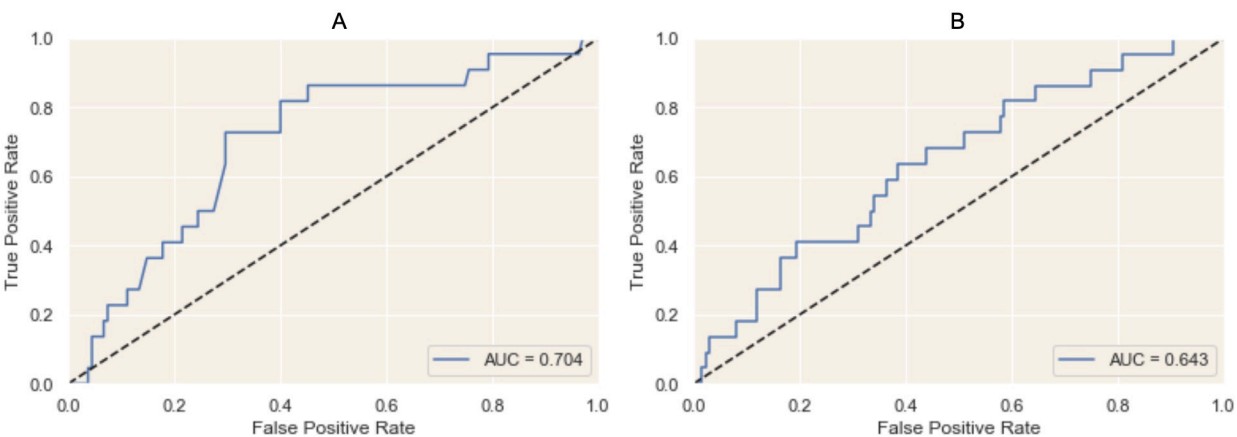

**Fig 3. Receiver operating characteristic for predictive frailty on in-clinic walk tests.** This figure represents the ability of A) at-home 6MWT data to predict in-clinic 6MWT and B) passive activity data to predict in-clinic 6MWT using a logistic regression model. The AUC of A) is 0.704 while the AUC of B) is 0.643 suggesting that passive data is nearly as accurate as an at-home 6MWT in predicting frailty as measured by an in-clinic 6MWT.

remote monitoring [14, 17–19]. Smartphones, with their increasing prevalence even among the elderly population, present an opportunity to passively collect data [20]. While prior research has studied passively collected step counts in the setting of cardiovascular health, we present novel results by comparing passive data to supervised, in-clinic 6MWT [21].

We replicate our previous findings that showed that smartphones are more accurate at measuring steps than distance during a clinic-based 6MWT [12]. Seventy-one percent of the variation in ground truth distance is explained by the step counts reported by the iPhone. When using both the step count and the distance measured, the iPhone can detect "frailty" with 90% sensitivity and 85% specificity. This shows that the iPhone can measure a 6MWT in the clinic nearly as well as the gold-standard supervised 6MWT.

In this study we also find that passively collected data on smartphones and wearables can provide clinically meaningful and actionable insights in patients with CVD. Namely, our predictive model can use data collected by the patient from the comfort of their homes to predict the result of a supervised in-clinic 6MWT with an AUC of 0.704. A home-based 6MWT is 83% sensitive and 60% specific for detecting whether a patient would classify as "frail" on an in-clinic 6MWT. Our results show that passively collected data is almost as predictive as a home-based 6MWT at predicting traditional, clinic-based 6MWT results (AUC 0.643).

Frailty has been shown to be an independent risk factor for adverse outcomes across surgical specialties [22]. Many tools such as the modified Frailty Index, Memorial Sloan Kettering–Frailty Index, geriatric assessment, and the Risk Analysis Index have been developed as more objective measures to supplant the "eyeball test." When considering an elective procedure, patients that are deemed "frail" would ideally be enrolled in a "prehabilitation" program to optimize their nutrition and functional status. In urgent cases, patients deemed "frail" would be advised to have lower risk, less invasive procedures. A better understanding of surgical risk enhances the shared decision-making required to respect patient preferences and quality-of-life considerations. Passive activity data collection on a smartphone could be viewed as "activity as a vital sign" and an excellent proxy for frailty with excellent negative predictive values.

Both analyses—comparing smartphone data from clinic to clinic ground truth, and comparing smartphone data from home to clinic ground truth—revealed that step counts are more reliable than distance in patients with CVD. The latter analysis was most accurate when using the maximum of either phone or watch steps, suggesting that the devices are undercounting

steps. We hypothesize that this may be due to shorter, shuffled steps in our population as compared to the population in which Apple trained the phone and watch step count algorithms.

The ability of the smartphone and smartwatch to measure a 6MWT appears to degrade when transitioning from the clinic to the home setting. However, this is more likely a reflection of different walk test conditions than the accuracy of the devices themselves. Compared to the clinic setting, the home setting suffers from no regimented course, day-to-day variability in activity, weather, and occupational demands, as well as less motivation to push oneself. This study shows that, while digital devices can accurately capture the 6MWT, transitioning from clinic-based to remote-based monitoring faces challenges beyond just technology.

Notably, despite the lower "tech literacy" of our patient population with 23% being smartphone naïve, we found 84% of the patients were able to complete the full study and adhere to regular instructions such as digital surveys and self-administered 6MWTs. Although devices were only turned on and had a cellular connection on 78% of total study days, on 93% of these days there was robust data collection (S1 Table in S1 Appendix). Dropout rate has previously been a barrier to the adoption of remote data as a reliable clinical tool. While our study suggests that remote monitoring may be reliable among veterans, it is possible that remote monitoring may not be as effective for other marginalized patient populations, particularly those with intermittent access to electricity and cellular service and those with high degrees of medical mistrust. Accessibility remains a major consideration for widespread implementation.

The main drawbacks of this study stem from its small sample size and lack of demographic diversity. That the study population was 99% male is a result of the study being implemented at a VA hospital. Owing to our small sample size, there were not enough interventions in the study cohort to apply regression analysis to predict surgical outcomes using passive activity data. We also only collected ~5 months of post-surgery data, but future studies with longer time horizons could capture longitudinal outcomes, such as 1, 2, 3 or 5-year morbidity and mortality data. Longitudinal data are also important to assess sustained patient engagement, which is known to decrease over time especially when there are fewer in-person visits [23].

Moreover, the small sample size did not afford the statistical power to perform a subgroup analysis on the different interventions studied (endovascular, open peripheral bypass, TAVR, CABG). With a larger sample size, the VascTrac research application could give us an understanding of whether surgical intervention results in a statistically significant change in daily physical activity, which would enable inferences into activities of daily living, quality of life, and frailty. Additionally, the participants may have been subject to the Hawthorne effect such that their awareness of being enrolled and monitored may have induced more activity than was their baseline. However, implementation of VascTrac monitoring outside of a research study could still induce this effect, and regardless of the patients' level of activity our results suggest that the VascTrac system is capable of accurately capturing passive activity data.

Adherence to the study protocol depended on a "high touch" approach from clinical coordinators, who called veterans who did not complete their weekly home-based 6MWT to remind them. While this high-touch approach would not be feasible for monitoring patients at the community level, our results show that passive activity data were almost as clinically informative as the home-based 6MWT. The former only require a patient to have their phone charged and with them, which represents a far more attainable adherence goal than a weekly app-based 6MWT.

## Conclusions

While the benefits of telemedicine and remote monitoring—convenience, low cost, improved data quality—have been postulated for some time, the COVID-19 pandemic has made

accelerated implementation a safety imperative. In this study, we showed that smart device-based measurements, including both a 6MWT and passively collected activity data, provide clinically accurate and meaningful insights about functional capacity in patients with CVD.

## Supporting information

**S1 Appendix.**
(DOCX)

**S1 File.**
(ZIP)

## Author Contributions

**Conceptualization:** Neil Rens, Neil Gandhi, Dasha Savage, Santiago Gutierrez, Oliver Aalami.

**Data curation:** Neil Rens, Neil Gandhi, Dasha Savage, Helle Nielsen-Bowles, Doran Triggs, Ghausia Ata, Julia Talgo, Oliver Aalami.

**Formal analysis:** Neil Rens, Neil Gandhi, Jonathan Mak, Jeddeo Paul, Drew Bent, Stephanie Liu.

**Funding acquisition:** Oliver Aalami.

**Investigation:** Neil Gandhi, Dasha Savage, Helle Nielsen-Bowles, Doran Triggs, Ghausia Ata, Oliver Aalami.

**Methodology:** Neil Rens, Neil Gandhi, Jonathan Mak, Jeddeo Paul, Drew Bent, Stephanie Liu, Santiago Gutierrez, Oliver Aalami.

**Project administration:** Neil Gandhi, Dasha Savage, Helle Nielsen-Bowles, Doran Triggs, Ghausia Ata, Julia Talgo, Oliver Aalami.

**Resources:** Helle Nielsen-Bowles, Oliver Aalami.

**Software:** Neil Rens, Jonathan Mak, Santiago Gutierrez, Oliver Aalami.

**Supervision:** Dasha Savage, Helle Nielsen-Bowles, Doran Triggs, Ghausia Ata, Oliver Aalami.

**Validation:** Neil Rens, Jonathan Mak, Jeddeo Paul.

**Visualization:** Neil Rens, Jonathan Mak, Jeddeo Paul.

**Writing – original draft:** Neil Rens, Neil Gandhi, Jonathan Mak, Oliver Aalami.

**Writing – review & editing:** Neil Rens, Neil Gandhi, Oliver Aalami.

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
