## [Decision Letter · Decision Letter 0]

31 Dec 2020

PONE-D-20-33554

Activity data from wearables as an indicator of functional capacity in patients with cardiovascular disease

PLOS ONE

Dear Dr. Rens,

Thank you for submitting your manuscript to PLOS ONE. After careful consideration, we feel that it has merit but does not fully meet PLOS ONE’s publication criteria as it currently stands. Therefore, we invite you to submit a revised version of the manuscript that addresses the points raised during the review process.

We look forward to receiving your revised manuscript.

Kind regards,

Chiara Lazzeri

Academic Editor

PLOS ONE

Journal Requirements:

"This study received funding from Apple Inc. and the Precision Health and Integrated Diagnostics Center at Stanford. None of the funding sources impacted the study design, data collection, analysis, decision to publish, or preparation of this manuscript.

https://www.apple.com

" ext-link-type="uri" xlink:type="simple">https://med.stanford.edu/phind.html"

We note that you received funding from a commercial source: Apple Inc.

Reviewers' comments:

Reviewer's Responses to Questions

**Comments to the Author**

1. Is the manuscript technically sound, and do the data support the conclusions?

Reviewer #1: Yes

Reviewer #2: Yes

2. Has the statistical analysis been performed appropriately and rigorously? 

Reviewer #1: I Don't Know

Reviewer #2: I Don't Know

3. Have the authors made all data underlying the findings in their manuscript fully available?

Reviewer #1: Yes

Reviewer #2: Yes

4. Is the manuscript presented in an intelligible fashion and written in standard English?

Reviewer #1: Yes

Reviewer #2: Yes

5. Review Comments to the Author

Reviewer #1: This is an interesting paper demonstrating the potential usefulness of remote monitoring of the performance status in cardiovascular patients. Although the protocol has some limitations (that were acknowledged by the Authors), I believe that the results are interesting and sound.

Just one remark: although the passive data were continuously recorded, it is not possible to exclude that the participation to the study protocol acted itself as a "Motivator", inducing patients to be somehow more active than usual.

Reviewer #2: The authors describe a study in which they monitored a cohort of preoperative/preprocedural cardiovascular disease patients smartphone data remotely to assess functional capacity and predict frailty

The authors presents a well written and creatively designed protocol to assess passive data collection from iphone technology with a measured comparison with in clinic 6 min walk testing.

Future question – validate longitudinal outcomes with activity data

Dropout rate – suggests a possible current limitation of remote data as a reliable clinical tool in all clinical settings – this may be a barrier to reliance on this emerging technology for certain potentially marginalized patient cohorts

Would be interesting to assess “frailty” assessment with other indicators of comorbidity and or a total burden of comorbidities

Please comment on the determination of “frailty” and resource allocation in the context of care and or prognostic information

It is important to comment on the long term prospects for sustained patient engagement and use when incorporating novel digital technology and data collection in patient care (home, outpatient, inpatient settings)

6. PLOS authors have the option to publish the peer review history of their article (what does this mean?). If published, this will include your full peer review and any attached files.

Reviewer #1: No

Reviewer #2: **Yes: **Ilan Kedan, MD MPH Smidt Cedars Sinai Heart Institute

---

## [Author Response · Author response to Decision Letter 0]

11 Feb 2021

Dear Reviewers, 

Thank you for your thoughtful reading of our manuscript entitled Activity data from wearables as an indicator of functional capacity in patients with cardiovascular disease. You highlighted some important considerations that we address below point-by-point. The actual edits are included both in the Manuscript as well as in the Response Letter file that was uploaded. Below, we only include references to the line number where the edits may be seen.

Reviewer #1: Although the passive data were continuously recorded, it is not possible to exclude that the participation to the study protocol acted itself as a "Motivator", inducing patients to be somehow more active than usual.

 Thank you for raising this important insight. We have included a few sentences in the discussion (line 268) addressing the fact that the study protocol itself may have been an activity motivator and therefore patient activity outside of a study protocol may differ.

Reviewer #2: Future question – validate longitudinal outcomes with activity data

 We wholeheartedly agree that future studies should assess longitudinal outcomes. We acknowledged that we did not have enough interventions in our study cohort to apply regression analysis to assess outcomes. We have also added the following language to the discussion of study limitations (line 258).

Reviewer #2: Dropout rate – suggests a possible current limitation of remote data as a reliable clinical tool in all clinical settings – this may be a barrier to reliance on this emerging technology for certain potentially marginalized patient cohorts

 You raise an important point about the potential for widespread adoption of remote monitoring. We have added additional discussion about the dropout rate in our study population and how that dropout rate may differ for other marginalized patient populations (line 248).

Reviewer #2: Would be interesting to assess “frailty” assessment with other indicators of comorbidity and or a total burden of comorbidities

Reviewer #2: Please comment on the determination of “frailty” and resource allocation in the context of care and or prognostic information

 Thank you for highlighting this question about the clinical utility of frailty. We have added a paragraph to the discussion to explain the clinical context for frailty and have added a citation for various other frailty measures. While other non-activity based measures of frailty use other indicators of comorbidity, our study did not have access to this data and therefore we could not compare frailty as predicted by remote monitoring these other frailty measures (line 216).

Reviewer #2: It is important to comment on the long term prospects for sustained patient engagement and use when incorporating novel digital technology and data collection in patient care (home, outpatient, inpatient settings)

 While we agree this is an important topic to discuss, given the 6-month time period of this study, we do not feel we have the data to extrapolate about sustaining patient engagement beyond 6 months. However, we have added a reference to a relevant study on patient engagement during a home-based exercise intervention monitored with a wearable (line 258).

Thank you for your consideration of our revised manuscript,

Neil Rens 

---

## [Editor Report · Decision Letter 1]

15 Feb 2021

Activity data from wearables as an indicator of functional capacity in patients with cardiovascular disease

PONE-D-20-33554R1

Dear Dr. Rens,

We’re pleased to inform you that your manuscript has been judged scientifically suitable for publication and will be formally accepted for publication once it meets all outstanding technical requirements.

Kind regards,

Chiara Lazzeri

Academic Editor

PLOS ONE
---

## [Editor Report · Acceptance letter]

1 Mar 2021

PONE-D-20-33554R1 

Activity data from wearables as an indicator of functional capacity in patients with cardiovascular disease 

Dear Dr. Rens:

I'm pleased to inform you that your manuscript has been deemed suitable for publication in PLOS ONE. Congratulations! Your manuscript is now with our production department. 

Kind regards, 

on behalf of

Dr. Chiara Lazzeri 

Academic Editor

PLOS ONE